# PDE-NetGen 1.0: from symbolic PDE representations of physical processes to trainable neural network representations.

Olivier Pannekoucke[1] and Ronan Fablet[2]

[1]INPT-ENM, UMR CNRS CNRM 3589, CERFACS, 42, av. G. Coriolis 31057 Toulouse, France
[2]IMT-Atlantic, UMR CNRS Lab-STICC, Brest, France

**Correspondence:** Olivier Pannekoucke (olivier.pannekoucke@meteo.fr)

**Abstract.** Bridging physics and deep learning is a topical challenge. While deep learning frameworks open avenues in physical science, the design of physically-consistent deep neural network architectures is an open issue. In the spirit of physics-informed NNs, *PDE-NetGen* package provides new means to automatically translate physical equations, given as PDEs, into neural network architectures. *PDE-NetGen* combines symbolic calculus and a neural network generator. The later exploits NN-based implementations of PDE solvers using Keras. With some knowledge of a problem, *PDE-NetGen* is a plug-and-play tool to generate physics-informed NN architectures. They provide computationally-efficient yet compact representations to address a variety of issues, including among others adjoint derivation, model calibration, forecasting, data assimilation as well as uncertainty quantification. As an illustration, the workflow is first presented for the 2D diffusion equation, then applied to the data-driven and physics-informed identification of uncertainty dynamics for the Burgers equation.

*Copyright statement.* TEXT

## 1 Introduction

Machine learning and deep learning receive a fast growing interest in geo-science to address open issues, including for instance sub-grid parmeterization,

A variety of learning architectures have shown their ability to encode the physics of a problem, especially deep learning schemes which typically involve millions of unknown parameters, while the theoretical reason of this success remains a key issue (Mallat, 2016). A recent research trend has involved the design of lighter neural network (NN) architectures, like ResNets with shared weights (He et al., 2016), while keeping similar learning performance. Interestingly, a ResNet can be understood as an implementation of a numerical time scheme solving a ODE/PDE (Ruthotto and Haber, 2019; Rousseau et al., 2019). Applications to learning PDEs from data have also been introduced *e.g.* PDE-Net (Long et al., 2017, 2018). These previous works emphasize the connection between the underlying physics and the NN architectures.

Designing or learning a NN representation for a given physical process remains a difficult issue. If the learning fails, it may be unclear to know how to improve the architecture of the neural network. Besides, it seems irrelevant to run computationally-

expensive numerical experiments on large-scale dataset to learn well-represented processes. The advection in fluid dynamics may be a typical example of such processes, which do not require complex non-linear data-driven representations. Overall, one would expect to accelerate and make more robust the learning process by combining, within the same NN architecture, the known physical equations with the unknown physics.

From the geoscience point of view, a key question is to bridge physical representations and neural network ones so that we can decompose both known and unknown equations according to the elementary computational units made available by state-of-the-art frameworks (e.g., keras, tensorflow). In other words, we aim to translate physical equations into the computational vocabulary available to neural networks. *PDE-NetGen* (Pannekoucke, 2020) addresses this issue for PDE representations, for which we regard convolutional layers as being similar to the stencil approach, which results from the discretization of the PDEs by using the finite-difference method (see *e.g.* Thomas (1995)). *PDE-NetGen* relies on two main components: (i) a computer algebra system, here Sympy (Meurer et al., 2017), used to handle the physical equations and discretize the associated spatial derivatives, (ii) a Keras network generator which automaticaly translate PDEs into neural network layers from these discretized forms. Note that code generator based on symbolic computation receives new interests to facilitate the design of numerical experiments see *e.g.* Louboutin et al. (2019). As an illustration, we consider in this paper the application of *PDE-NetGen* to the identification of closure terms.

The paper is organized as follows. In the next section, we detail the proposed neural network generator, with an illustration of the workflow on a diffusion equation. In section 3, we present the numerical integration of the neural network implementation of the diffusion equation then an application to the data-driven identification of the closure of Burgers equation. Conclusion and perspective are given in section 4

## 2    Neural Network Generatation from symbolic PDEs

Introducing physics in the design of neural network topology is challenging since physical processes can rely on very different partial derivative equations, *e.g.* eigenvalue problems for waves or constrained evolution equations in fluid dynamics under iso-volumetric assumption. The neural network code generator presented here focuses on physical processes given as evolution equations which writes

$$\partial_t u = \mathcal{M}(u, \partial^\alpha u), \tag{1}$$

where $u$ denotes either a scalar field or multivariate fields, $\partial^\alpha u$ denotes partial derivatives with respect to spatial coordinates, and $\mathcal{M}$ is the generator of the dynamics. At first glance, this situation excludes diagnostic equation as encountered in geophysics, like balance equations: each equation has to be the evolution equation of a prognostic variable. *PDE-NetGen* incorporates a way to solve diagnostic equation, this will be shown in the example detailed in Section 3.2.

We first explain how the derivatives are embedded into NN layers, then we detail the workflow of *PDE-NetGen* for a simple example.

## 2.1 Introducing physical knowledge in the design of a NN topology

Since the NN generator is designed for evolution equations, the core of the generator is the automatic translation of partial derivatives with respect to spatial coordinates into layers. The correspondence between the finite-difference discretization and the convolutional layer give a practical way to translate a PDE into a NN (Cai et al., 2012; Dong et al., 2017; Long et al., 2017).

The finite-difference method remains to replace the derivative of a function by a fraction that only depends on the value of the function (see *e.g.* Thomas (1995)). For instance, the finite-difference method applied on a second order partial derivative $\partial_x^2 u$, for $u(t,x)$ on a 1D domain, leads to approximate the derivative by

$$\partial_x^2 u(t,x) \approx \mathcal{F}_x^2 u(t,x), \tag{2}$$

with

$$\mathcal{F}_x^2 u(t,x) = \frac{u(t, x+\delta x) + u(t, x-\delta x) - 2u(t,x)}{\delta x^2}, \tag{3}$$

where $\delta x$ stands for the discretization space step. Here the spatial derivative is replaced by a fraction that only depends on the values of $u$ at the time $t$ and points $x - \delta x$, $x$, $x + \delta x$. This makes appear a kernel stencil $k = [1/\delta x^2, -2/\delta x^2, 1/\delta x^2]$ that can be used in a 1D convolution layer with a linear activation function and without bias. A similar routine applies for 2D and 3D geometries. *PDE-NetGen* relies on the computer algebra system *sympy* (Meurer et al., 2017) to compute the stencil as well as to handle symbolic expressions.

In *PDE-NetGen*, the finite-difference implementation appears as a linear operator $\mathcal{F}$ which approximates any partial derivative from the values on a regular grid. In particular, the finite difference $\mathcal{F}_x^\alpha u(t,x)$ of any partial derivative $\partial_x^\alpha u(t,x)$ of order $\alpha$, is computed from the grid points $\{x \pm (2i+1)\delta x\}_{i \in [0,p]}$ when $\alpha = 2p+1$ is odd and $\{x \pm i\delta x\}_{i \in [0,p]}$ when $\alpha = 2p$ is even. This approximation is consistent at the second order *i.e.* $\mathcal{F}_x^\alpha u \underset{0}{=} \partial_x^\alpha u + \mathcal{O}(\delta x^2)$, where $\mathcal{O}$ is the Landau's big O notation: for any $f$, the notation $f(\delta x) \underset{0}{=} \mathcal{O}(\delta x^2)$ means that $\lim_{\delta x \to 0} \frac{f(\delta x)}{\delta x^2}$ is finite. The operator $\mathcal{F}$ behaves partially as the partial derivative operator $\partial$: $\mathcal{F}$ is commutative with respect to independent coordinates *i.e.* in a 2D domain for coordinates $(x,y)$ we have $\mathcal{F}_x \circ \mathcal{F}_y = \mathcal{F}_y \circ \mathcal{F}_x$, where $\circ$ denotes the operator composition, and this applies at any order *e.g.* $\mathcal{F}_{xxy}^3 = \mathcal{F}_x^2 \circ \mathcal{F}_y$ (but $\mathcal{F}_x^2 \neq \mathcal{F}_x \circ \mathcal{F}_x$). Hence, the finite difference of a derivative with respect to multiple coordinate, is computed sequentially from the iterative discretization along each coordinate, and this approximation is consistent at the second order. Note that we chose to design *PDE-NetGen* considering the finite-difference method, but alternatives using automatic differentiation can be considered as introduced by Raissi (2018) who used TensorFlow for the computation of derivative.

Then, the time integration can be implemented either by a solver or by a ResNet architecture of a given time scheme *e.g.* an Euler scheme or a fourth order Runge-Kutta (RK4) scheme (Fablet et al., 2017).

These two components, namely the translation of partial derivatives into NN layers and a ResNet implementation of the time integration, are the building blocks of the proposed NN topology generator as examplified in the next Section.

```python
from sympy import Function, symbols, Derivative
from pdenetgen import Eq, NNModelBuilder

# Defines the diffusion equation using sympy
t, x, y = symbols('t x y')
u = Function('u')(t,x,y)
kappa11 = Function('\\kappa_{11}')(x,y)
kappa12 = Function('\\kappa_{12}')(x,y)
kappa22 = Function('\\kappa_{22}')(x,y)

diffusion_2D = Eq(Derivative(u,t),
  Derivative(kappa11*Derivative(u,x)+
            kappa12*Derivative(u,y),x)+
  Derivative(kappa12*Derivative(u,x)+
            kappa22*Derivative(u,y),y)).doit()

# Defines the neural network code generator
diffusion_nn_builder = NNModelBuilder(diffusion_2D,
        class_name="NNDiffusion2DHeterogeneous")

# Renders the neural network code
exec(diffusion_nn_builder.code)

# Create a 2D Diffusion model
diffusion_model = NNDiffusion2DHeterogeneous()
```

**Figure 1.** Neural Network generator for a heterogeneous 2D diffusion equation

## 2.2 Workflow of the NN representation generator

We now present the workflow for the NN generator given a symbolic PDE using the heterogeneous 2D diffusion equation as a testbed:

$$\partial_t u = \nabla \cdot (\boldsymbol{\kappa} \nabla u), \tag{4}$$

where $\boldsymbol{\kappa}(x,y) = [\kappa_{ij}(x,y)]_{(i,j)\in[1,2]\times[1,2]}$ is a field of $2 \times 2$ tensors $((x,y)$ are the spatial coordinates) and whose python implementation is detailed in Fig. 1.

Starting from a list of coupled evolution equations given as a PDE, a first preprocessing of the system determines the prognostic functions, the constant functions, the exogenous functions and the constants. The exogenous functions are the functions which depends on time and space, but whose evolution is not described by the system of evolution equations. For instance, a forcing term in a dynamics is an exogenous function.

For the diffusion equation Eq. (4), the dynamics is represented in *sympy* using *Function*, *Symbol* and *Derivative* classes. The dynamics is defined as an equation using the *Eq* class of *PDE-NetGen*, which inherits from the *Eq* class of *sympy* with additional facilities (see the implementation in Fig. 1 for additional details).

The core of the NN generator is given by the *NNModelBuilder* class. This class first preprocesses the system of evolution equations and translates the system into a python NN model.

```
# Example of computation of a derivative
kernel_Du_x_o1 = np.asarray([[0.0,-1/(2*self.dx[self.coordinates.index('x')]),0.0],
 [0.0,0.0,0.0],
 [0.0,1/(2*self.dx[self.coordinates.index('x')]),0.0]]).reshape((3, 3)+(1,1))
Du_x_o1 = DerivativeFactory((3, 3),kernel=kernel_Du_x_o1,name='Du_x_o1')(u)

# Computation of trend_u
mul_0 = keras.layers.multiply([Dkappa_11_x_o1,Du_x_o1],name='MulLayer_0')
mul_1 = keras.layers.multiply([Dkappa_12_x_o1,Du_y_o1],name='MulLayer_1')
mul_2 = keras.layers.multiply([Dkappa_12_y_o1,Du_x_o1],name='MulLayer_2')
mul_3 = keras.layers.multiply([Dkappa_22_y_o1,Du_y_o1],name='MulLayer_3')
mul_4 = keras.layers.multiply([Du_x_o2,kappa_11],name='MulLayer_4')
mul_5 = keras.layers.multiply([Du_y_o2,kappa_22],name='MulLayer_5')
mul_6 = keras.layers.multiply([Du_x_o1_y_o1,kappa_12],name='MulLayer_6')
sc_mul_0 = keras.layers.Lambda(lambda x: 2.0*x,name='ScalarMulLayer_0')(mul_6)
trend_u = keras.layers.add([mul_0,mul_1,mul_2,mul_3,mul_4,mul_5,sc_mul_0],name='AddLayer_0')
```

**Figure 2.** Part of the python code of the *NNDiffusion2DHeterogeneous* class which implements the diffusion equation Eq. (4) as a neural-network by using Keras (only one derivative is explicitly given, for the sake of simplicity)

The preprocessing of the diffusion equation Eq. (4) presents a single prognostic function, $u$, and three constant functions $\kappa_{11}, \kappa_{12}$ and $\kappa_{22}$. There is no exogenous function for this example. During the preprocessing, the coordinate system of each function is diagnosed such that we may determine the dimension of the problem. For the diffusion equation Eq. (4), since the function $u(t,x,y)$ is a function of $(x,y)$ the geometry is two-dimensional. In the current version of *PDE-NetGen*, only periodic boundaries are considered. The specific *DerivativeFactory* class ensures the periodic extension of the domain, then the computation of the derivative by using CNN and finally the crop of the extended domain to return to the initial domain. Other boundaries could also be implemented and might be investigated in future developments.

All partial derivatives with respect to spatial coordinates are detected and then replaced by an intermediate variable in the system of evolution equations. The resulting system is assumed to be algebraic, which means that it only contains addition, subtraction, multiplication and exponentiation (with at most a real). For each evolution equation, the abstract syntax tree is translated into a sequence of layers which can be automatically converted into NN layers in a given NN framework. For the current version of *PDE-NetGen*, we consider *Keras* (Chollet, 2018). An example of the implementation in *Keras* is shown in Fig. 2: a first part of the code is used to compute all the derivatives using Conv layers of *Keras*, then *Keras* layers are used to implement the algebraic equation which represents the trend $\partial_t u$ of the diffusion equation Eq. (4).

At the end, a python code is rendered from templates by using the *jinja2* package. The reason why templates are used is to facilitate the saving of the code in python modules and the modification of the code by the experimenter. Runtime computation of the class could be considered, but this is not implemented in the current version of *PDE-NetGen*. For the diffusion equation Eq. (4), when run, the code rendered from the *NNModelBuilder* class creates the *NNDiffusion2DHeterognous* class. Following the class diagram Fig. 3, the *NNDiffusion2DHeterogeneous* class inherits from a *Model* class which implements the time evolution of an evolution dynamics by incorporating a time-scheme. Here several time-schemes are implemented, namely an explicit Euler scheme, a second and a fourth order Runge-Kutta scheme.

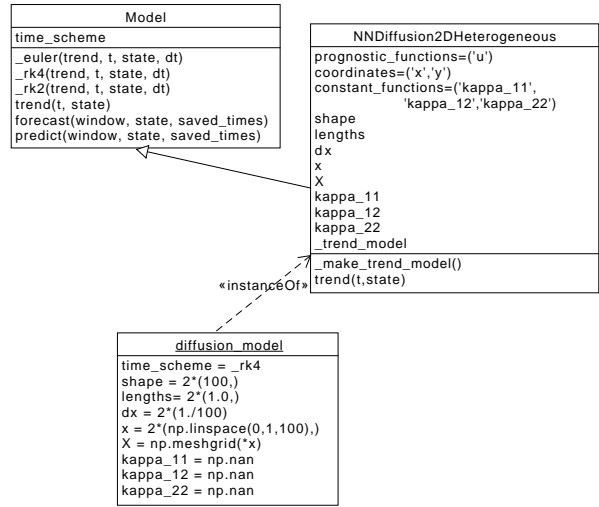

**Figure 3.** UML class diagram showing the interaction between the *Model* and the *NNDiffusion2DHeterogeneous* classes, and the resulting instance *diffusion_model* corresponding to the numerical computation of the diffusion equation Eq. (4).

## 3 Applications of *PDE-NetGen*

Two applications are now considered. First we validate the NN generator on a known physical problem: the diffusion equation Eq. (4) detailed in the previous section. Then, we tackle a situation where a part of the physics remains unknown, showing the benefit of merging the known physics in the learning of the unknown processes.

### 3.1 Application to the diffusion equation

In the python implementation Fig. 1, *diffusion_model* is an instance of the *NNDiffusion2DHeterogeneous* class, which numerically solves the diffusion equation Eq. (4) over a 2D domain, defined by default as the periodic domain $[0,1) \times [0,1)$ discretized by 100 points along each directions, so that $dx = dy = 1.0/100$.

The time integration of the diffusion equation is shown in Fig. 4. For this numerical experiment, the heterogeneous tensor field of diffusion tensors $\boldsymbol{\kappa}(x,y)$ is set as rotations of the diagonal tensor $(l_x^2/\tau, l_y^2/\tau)$ defined from the length-scales $l_x = 10\,dx$, $l_y = 5\,dy$ and the time-scale $\tau = 1.0$, and with the rotation angles $\theta(x,y) = \frac{\pi}{3}\cos(k_x x + k_y y)$ where $(k_x, k_y) = 2\pi(2,3)$. The time step for the simulation is $dt = \tau Min(dx^2/lx^2, dy/ly^2)/6 \approx 1.66 \quad 10^{-3}$. The numerical integration is computed by using a fourth-order Runge-Kutta scheme. The initial condition of the simulation is given by a Dirac Fig. 4 (a). In order to validate the solution obtained from the generated neural network, we compare the integration with the one of the finite-difference discretization of Eq. (4),

$$\partial_t u = \mathcal{F}_{x^i}(\kappa_{ij})\mathcal{F}_{x^j}(u) + \kappa_{ij}\mathcal{F}^2_{x^i x^j}(u), \tag{5}$$

where $\mathcal{F}$ is the operator described in Section 2.1, and whose numerical result is shown in Fig. 4 (b).

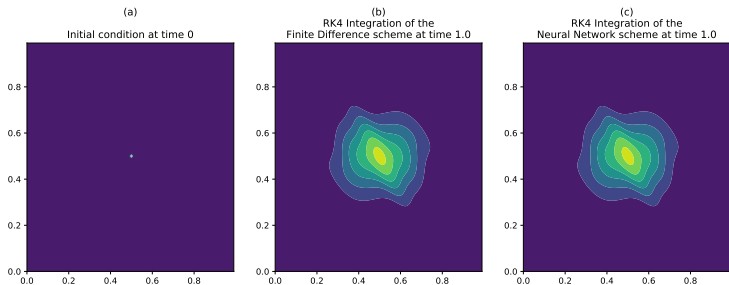

**Figure 4.** Starting from a Dirac (panel a), the diffusion equation Eq. (4) is integrated from 0 to 1 by using a fourth-order Runge-Kutta time scheme. The results obtained from the time integration of the finite-difference implementation Eq. (5) (panel b) and of the generated NN representation (panel c) are similar.

The heterogeneity of the diffusion tensors makes appear an anisotropic diffusion of the Dirac (see Fig. 4 (b)), which is perfectly reproduced by the result obtained from the integration of the generated neural network, shown in Fig. 4 (c). At a quantitative level, the $l^2$ distance between the both solutions is $10^{-5}$ (with $dt = 1.6 \quad 10^{-3}$). This validates the ability of the NN generator *PDE-NetGen* to compute the dynamics of a given physical evolution equation.

The next section illustrates the situation where only a part of the dynamics is known, while the remaining physics is learned from the data.

## 3.2 Application to the data-driven identification of stochastic representations

As an illustration of the *PDE-NetGen* package, we consider a problem encountered in uncertainty prediction: the parametric Kalman filter (PKF) (Pannekoucke et al., 2016, 2018). For a detailed presentation and discussion of uncertainty prediction issues in geophysical dynamics, we may refer the reader to Le Maître and Knio (2010). Here, we briefly introduce basic elements for the self-consistency of the example.

The idea of the PKF is to mimic the dynamics of the covariance-error matrices all along the analysis and the forecast cycle of the data assimilation in a Kalman setting (Kalman filter equations for the uncertainty). It relies on the approximation of the true covariance matrices by some parametric covariance model. When considering a covariance model based on a diffusion equation, the parameters are the variance $V$ and the local diffusion tensor $\nu$. Therefore, the dynamics of the covariance-error matrices along the data assimilation cycles is deduced from the dynamics of the variance and of the diffusion tensors. In place of the full covariance evolution this dramatically reduces the dynamics to the one of few parameters.

For the non-linear advection-diffusion equation, known as the Burgers equation,

$$\partial_t u + u \partial_x u = \kappa \partial_x^2 u, \tag{6}$$

the dynamics of the variance $V_u$ and the diffusion tensor $\boldsymbol{\nu}_u = [\nu_{u,xx}]$ (which is featured by a single field $\nu_{u,xx}$), writes (Pannekoucke et al., 2018)

$$
\begin{cases}
\frac{\partial}{\partial t} u &= \kappa \frac{\partial^2}{\partial x^2} u - u \frac{\partial}{\partial x} u - \frac{\frac{\partial}{\partial x} V_u}{2} \\[1mm]
\frac{\partial}{\partial t} V_u &= -\frac{\kappa V_u}{\nu_{u,xx}} + \kappa \frac{\partial^2}{\partial x^2} V_u - \frac{\kappa \left(\frac{\partial}{\partial x} V_u\right)^2}{2 V_u} \\[1mm]
& \quad - u \frac{\partial}{\partial x} V_u - 2 V_u \frac{\partial}{\partial x} u \\[1mm]
\frac{\partial}{\partial t} \nu_{u,xx} &= 4 \kappa \nu_{u,xx}{}^2 \, \mathbb{E}\left[\varepsilon_u \frac{\partial^4}{\partial x^4} \varepsilon_u\right] \\[1mm]
& \quad - 3\kappa \frac{\partial^2}{\partial x^2} \nu_{u,xx} - \kappa + \frac{6\kappa \left(\frac{\partial}{\partial x}\nu_{u,xx}\right)^2}{\nu_{u,xx}} \\[1mm]
& \quad - \frac{2\kappa \nu_{u,xx} \frac{\partial^2}{\partial x^2} V_u}{V_u} + \frac{\kappa \frac{\partial}{\partial x} V_u \frac{\partial}{\partial x} \nu_{u,xx}}{V_u} + \\[1mm]
& \quad \frac{2\kappa \nu_{u,xx}\left(\frac{\partial}{\partial x}V_u\right)^2}{V_u{}^2} - u \frac{\partial}{\partial x} \nu_{u,xx} + \\[1mm]
& \quad 2 \nu_{u,xx} \, \frac{\partial}{\partial x} u
\end{cases}
\tag{7}
$$

where $\mathbb{E}\left[\cdot\right]$ denotes the expectation operator. For the sake of simplicity, in this system of PDEs, $u$ denotes the expectation of the random field and not the random field itself as in (Eq. (6)).

In this system of PDEs, the term $\mathbb{E}\left[\varepsilon_u \frac{\partial^4}{\partial x^4} \varepsilon_u\right]$ can not be determined from the known quantities $u, V_u$ and $\nu_{u,xx}$. This brings up a problem of closure, *i.e.* determinining the unknown term as a function of the known quantities. A naive assumption would be to consider a zero closure ($closure(t,x) = 0$). However, while the tangent-linear evolution of the perturbations along the Burgers dynamics is stable, the dynamics of the diffusion coefficient $\nu_{u,xx}$ would lead to unstable dynamics as the coefficient of the second order term $-3\kappa \frac{\partial^2}{\partial x^2} \nu_{u,xx}$ is negative. This stresses further the importance of the unknown term to successfully predict the uncertainty.

Within a data-driven framework, one would typically explore a direct identification of the dynamics of diffusion coefficient $\nu_{u,xx}$. Here, we exploit *PDE-NetGen* to fully exploit the known physics and focus on the data-driven identification of the unknown term $\mathbb{E}\left[\varepsilon_u \frac{\partial^4}{\partial x^4} \varepsilon_u\right]$ in the system of equations Eq. (7). It comes to replace term $\mathbb{E}\left[\varepsilon_u \frac{\partial^4}{\partial x^4} \varepsilon_u\right]$ in Eq. (7) by an exogenous function *closure(t,x)* and then to follow the workflow detailed in Section 2.2.

The unknown closure function is represented by a neural network (a Keras model) which implements the expansion

$$
closure(t,x) \sim a \frac{\frac{\partial^2}{\partial x^2} \nu_{u,xx}(t,x)}{\nu_{u,xx}{}^2(t,x)} + b \frac{1}{\nu_{u,xx}{}^2(t,x)} + c \frac{\left(\frac{\partial}{\partial x}\nu_{u,xx}(t,x)\right)^2}{\nu_{u,xx}{}^3(t,x)}
\tag{8}
$$

where $(a,b,c)$ are unknown and where the partial derivatives are computed from convolution layers, as described in Section 2. This expression is similar to a dictionary of possible terms as in Rudy et al. (2017) and it is inspired from an arbitrary theoretically-designed closure for this problem where $(a,b,c) = (1, \frac{3}{4}, -2)$ (see Appendix A for details). In the NN implementation of the exogenous function modeled as Eq. (8), each of the unknown coefficients $(a,b,c)$ are implemented as a 1D convolutional layer, with a linear activation function and without bias. Note that the estimated parameters $(a,b,c)$ could be different from the one of the theoretical closure: while the theoretical closure can give some clues for the design of the unknown term, this closure is not the truth which is unknown (see Appendix A).

The above approach, which consists in constructing an exogenous function given by a NN to be determined, may seem tedious for an experimenter who would not be accustomed to NNs. Fortunately, we have considered an alternative in *PDE-*

**Introduction of the closure ine the PKF dynamics**

```python
from pdenetgen import TrainableScalar

# Set the closure by using TrainableScalar
a, b, c = [TrainableScalar(l) for l in 'abc']
closure_proposal = a*Derivative(nu,x,2)/nu**Integer(2)+b*1/nu**Integer(2)+\
                c*Derivative(nu,x)**2/nu**Integer(3)
display(closure_proposal)
```

$$a\frac{\frac{\partial^2}{\partial x^2}\,v_{u,xx}\,(t,x)}{v_{u,xx}{}^2\,(t,x)} + \frac{b}{v_{u,xx}{}^2\,(t,x)} + \frac{c\left(\frac{\partial}{\partial x}\,v_{u,xx}\,(t,x)\right)^2}{v_{u,xx}{}^3\,(t,x)}$$

```python
# Replace the closure(t,x) by the proposed closure
pkf_dynamics[2] = pkf_dynamics[2].subs(Function('closure')(t,x),closure_proposal)

# Generate the NN code leading to the ClosedPKFBurgers class.
exec(NNModelBuilder(pkf_dynamics,'ClosedPKFBurgers').code)
```

**Sample of code generated to define the ClosedPKFBurgers class**

```python
[..]
pow_21 = keras.layers.multiply([div_17,div_17,] ,name='PowLayer_21')
mul_28 = keras.layers.multiply([pow_21,Dnu_u_xx_x_o2],name='MulLayer_28')
train_scalar_9 = TrainableScalarLayerFactory(input_shape=mul_28.shape, name='TrainableScalar_a',
            init_value=0,use_bias=False,mean=0.0,stddev=1.0,seed=None,wl2=None)(mul_28)
                #TrainableScalar name: 'a'
add_8 = keras.layers.add([train_scalar_7,train_scalar_8,train_scalar_9],name='AddLayer_8')
mul_26 = keras.layers.multiply([pow_17,add_8],name='MulLayer_26')
[..]
```

**Figure 5.** Implementation of the closure, by defining each unknown quantity as an instance of the class *TrainableScalar*, and the resulting generated NN code. This is a part of code avaibale in the Jupyter notebook given as example in the package *PDE-NetGen*.

*NetGen* that can be used in the particular case where candidates for a closure take the form of an expression with partial derivatives, as it is the case for Eq. (8). An example of implementation is shown in Fig. 5 where *pkf_dynamics* stands for the system of equations Eq. (7). The unkown closure function is replaced by the proposal of closure Eq. (8) where each unknown quantity $(a,b,c)$ is declared as an instance of the class *TrainableScalar*. Then, the NN is generated producing the class *ClosedPKFBurgers* whose an instance is ready for training. In the generated code, each instance of the *TrainableScalar* class is translated as a specific layer, *TrainableScalarLayerFactory*, equivalent to the above mentioned convolution layer, and whose parameter can be trainable. For instance, the trainable scalar $a$ is implemented by the line *train_scalar_9*. Note that the layer *TrainableScalarLayerFactory* can be used for 1D, 2D or 3D domains. In this example, the proposal for closure has been defined at a symbolic level, without additional exogenous NN.

An example of implementation for the exogenous NN and for the Trainable layers are provided in the package *PDE-NetGen* as Jupyter notebooks, for the case of the Burgers equation.

For the numerical experiment, the Burgers equation is solved on a one-dimensional periodic domain of length 1, discretized in 241 points. The time step is $dt = 0.002$, and the dynamics is computed over 500 time steps so to integrate from $t = 0$ to $t = 1.0$. The coefficient of the physical diffusion is set to $\kappa = 0.0025$. The numerical setting considered for the learning is the tangent-linear regime described in Pannekoucke et al. (2018) where the initial uncertainty is small and whose results are shown in their Fig. 4(a), Fig. 5(a) and Fig. 6(a).

```python
def make_time_scheme(dt, trend):
    """ Implementation of an RK4 with Keras """
    import keras

    state = keras.layers.Input(shape = trend.input_shape[1:])

    # k1
    k1 = trend(state)
    # k2
    _tmp_1 = keras.layers.Lambda(lambda x : 0.5*dt*x)(k1)
    input_k2 = keras.layers.add([state,_tmp_1])
    k2 = trend(input_k2)
    # k3
    _tmp_2 = keras.layers.Lambda(lambda x : 0.5*dt*x)(k2)
    input_k3 = keras.layers.add([state,_tmp_2])
    k3 = trend(input_k3)
    # k4
    _tmp_3 = keras.layers.Lambda(lambda x : dt*x)(k3)
    input_k4 = keras.layers.add([state,_tmp_3])
    k4 = trend(input_k4)

    # output
    # k2+k3
    add_k2_k3 = keras.layers.add([k2,k3])
    add_k2_k3_mul2 = keras.layers.Lambda(lambda x:2.*x)(add_k2_k3)
    # Add k1,k4
    _sum = keras.layers.add([k1,add_k2_k3_mul2,k4])
    # *dt
    _sc_mul = keras.layers.Lambda(lambda x:dt/6.*x)(_sum)
    output = keras.layers.add([state, _sc_mul])

    time_scheme = keras.models.Model(inputs =[state],
                                     outputs=[output])
    return time_scheme
```

**Figure 6.** Example of a Keras implementation for a RK4 time-scheme: given time-step $dt$ and a Keras model *trend* of the dynamics, the function *make_time_scheme* returns a Keras model implementing a RK4.

To train the parameters $(a, b, c)$ in Eq. (8), we build a training dataset from an ensemble prediction method where each member is a numerical solution of the Burgers equation. The numerical code for the Burgers equation derives from *PDE-NetGen* applied on the symbolic dynamics Eq. (6). Using this numerical code, we generate a training dataset composed of $400$ ensemble simulations of $501$ time steps, where each each ensemble contains $400$ members. For each ensemble forecast, we estimate the mean, variance $V_u$ and diffusion tensor $\nu_u$. Here, we focus on the development of the front where we expect the unknown term to be of key importance and keep for training purposes the last $100$ time-steps of each ensemble forecast. For the training only, the RK4 time-scheme is computed as the ResNet implementation given in Fig. 6, so to provide the end-to-end NN implementation of the dynamics.

The resulting dataset involves $40000$ samples. To train the learnable parameters $(a, b, c)$, we minimize the one-step ahead prediction loss for the diffusion tensor $\nu_u$. We use ADAM optimizer (Kingma and Ba, 2014) and a batch size of 32. Using an initial learning rate of $0.1$, the training converges within 3 outer loops of 30 epochs with a geometrical decay of the learning rate by a factor of $1/10$ after each outer loop. The coefficients resulting from the training over 10 runs are $(a, b, c) = (0.93, 0.75, -1.80) \pm (5.1 \ 10^{-5}, 3.6 \ 10^{-4}, 2.7 \ 10^{-4})$.

Figure 7 compares the estimation from a large ensemble of $1000$ members (top panels) with the results of the trained closed PKF dynamics (bottom panels). Both the ensemble and PKF means (a1) and (b1) clearly show a front which emerges from the smooth initial condition and located near $x = 0.75$ at time 1.. The variance fields (a2) and (b2) illustrate the vanishing

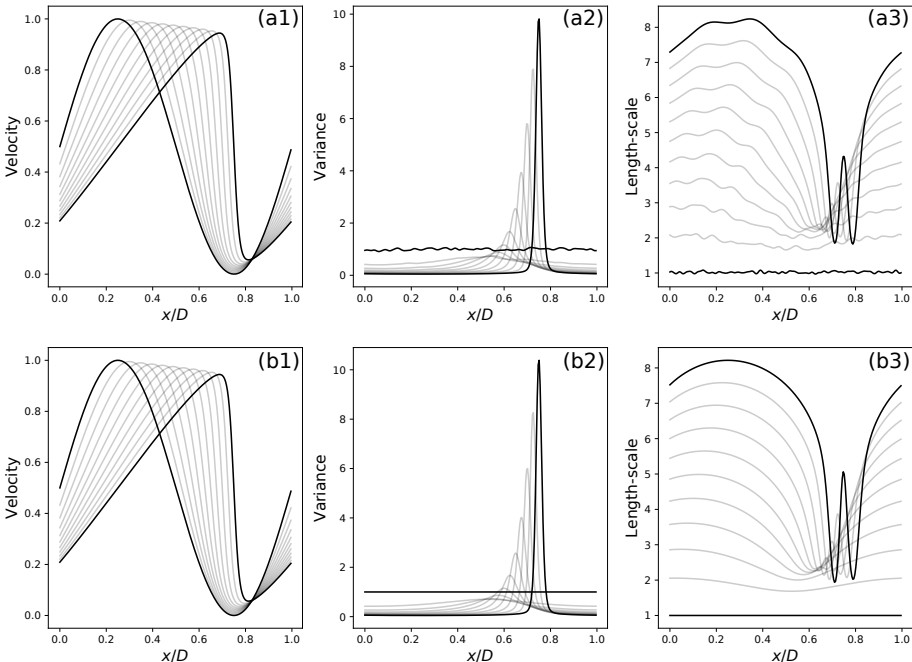

**Figure 7.** Uncertainty estimated from a large ensemble of 1000 members (a) with the expectation $\mathbb{E}[u]$ (a1), variance $V_u$ (a2) and the length-scale (defined from the diffusion coefficient by $\sqrt{0.5\,\nu_{u,xx}}$) (a3) ; and the uncertainty predicted from the PKF evolution equations closed from the data (b), where the same statistics are shown in (b1), (b2) and (b3). The fields are represented only for time $t = 0, 0.2, 0.4, 0.6, 0.8, 1$

of the variance due to the physical diffusion (the $\kappa$ term in Eq. (6)) and the emergence of a peak of uncertainty which is
215  related to the uncertainty of the front position. Instead of the diffusion $\nu_{u,xx}$, panels (a3) and (b3) show the evolution of the
correlation length-scale defined as $\sqrt{0.5\,\nu_{u,xx}}$, which has the physical dimension of a length. Both panels show the increase of
the length-scale due to the physical diffusion, except in the vicinity of the front where an oscillation occurs, which is related
to the inflexion point of the front. While the magnitude of the oscillation predicted by the PKF (b3) is slightly larger than the
estimation from the large ensemble reference (a3), the pattern is well predicted by the PKF. Besides, the parametric form of the
220  PKF does not involve local variabilities due to the finite size of the ensemble, which may be observed in panel (a3). Overall,
these experiments support the relevance of the closure Eq. (8) learned from the data to capture the uncertainty associated with
Burgers' dynamics.

### 3.3 Discussion on the choice of a closure

In the Burgers' dynamics, an a priori knowledge was introduced to propose a NN implementing the closure Eq. (8).
225  In the general case, the choice of the terms to be introduced in the closure may be guided by known physical properties
that need to be verified by the system. For example, conservation or symmetries properties that leave the system invariant
can guide in proposing possible terms For the Burgers' dynamics, $\nu_{u,xx}$ has the dimension of a length squared, $[L^2]$, and

$\mathbb{E}\left[\varepsilon_{\mathrm{u}}\frac{\partial^4}{\partial x^4}\varepsilon_{\mathrm{u}}\right]$ is of dimension $[L^{-4}]$. Thus, the terms considered in Eq. (8) are among the simplest ones which fullfill the expected dimensionality of $[L^{-4}]$. Symbolic computation may here help the design of such physical parameterizations in more general cases.

When no priors are available, one may consider modeling the closure using state-of-the-art deep neural network architectures which have shown impressive prediction performance, *e.g.* CNNs, ResNets (Zagoruyko and Komodakis, 2016; Raissi, 2018).

The aim of the illustration proposed for the Burgers' dynamics is not to introduce a deep learning architecture for the closure, but to facilitate the construction of a deep learning architecture taking into account the known physics: the focus is on the hybridation between physics and machine learning. Though the closure itself may not result in a deep architecture, the overall generated model leads to a deep architecture. For instance, the implementation using the exogenous NN use around 75 layers while the implementation based on the class *TrainableScalar* use 73 layers (we save the calculation of the derivatives that appear in Eq. (8), while they are computed twice when using the exogenous NN), with several convolutional layers among them. For other problems, there would be no other choice than considering a deep neural network, for instance using multiple ResNet blocks, normalization, and so on, or architectures inspired from recent studies on closure modeling (*e.g.* , Bolton and Zanna (2019)). Such architectures can be plugged in *PDE-NetGen* as an exogenous neural network.

## 4 Conclusions

We have introduced a neural network generator *PDE-NetGen*, which provides new means to bridge physical priors given as symbolic PDEs and learning-based NN frameworks. This package derives and implements a finite-difference version of a system of evolution equations, where the derivative operators are replaced by appropriate convolutional layers including the boundary conditions. The package has been developed in python using the symbolic mathematics library *sympy* and *keras*.

We have illustrated the usefulness of *PDE-NetGen* through two applications: a neural-network implementation of a 2D heterogeneous diffusion equation and the uncertainty prediction in the Burgers equation. The later involves unknown closure terms, which are learned from data using the proposed neural-network framework. Both illustrations show the potential of such an approach, which could be useful for improving the training in complex application by taking into account the physics of the problem.

This work opens new avenues to make the most of existing physical knowledge and of recent advances in data-driven settings, and more particularly neural networks, for geophysical applications. This includes a wide range of applications, where such physically-consistent neural network frameworks could either lead to the reduction of the computational cost (e.g., GPU implementation embedded in deep learning frameworks) or provide new numerical tools to derive key operators (e.g., adjoint operator using automatic differentiation). Besides, these neural network representations also offer new means to complement known physics with the data-driven calibration of unknown terms. This is regarded as key to advance the state-of-the-art for the simulation, forecasting and reconstruction of geophysical dynamics through model-data-coupled frameworks.

*Code availability.* The *PDE-NetGen* package is free and open source. It is distributed under the CeCILL-B free software license. The source code is provided through a GitHub repository https://github.com/opannekoucke/pdenetgen. A snapshot of *PDE-NetGen* 1.0 is available at https://doi.org/10.5281/zenodo.3891101 (Pannekoucke, 2020)

## Appendix A: Local Gaussian closure

For self-consistency, we detail how the theoretical closure is obtained (Pannekoucke et al., 2018).

It can be shown that $\mathbb{E}\left[\varepsilon_u \partial_x^4 \varepsilon_u\right] = \mathbb{E}\left[\left(\partial_x^2 \varepsilon_u\right)^2\right] - 2\partial_x^2 g_u$ where $g_u = \frac{1}{2\nu_u}$ is the so-called metric tensor that is a scalar field in 1D. When the correlation function $\rho(x, x+\delta x) = \mathbb{E}\left[\varepsilon(x)\varepsilon(x+\delta x)\right]$ is a homogeneous Gaussian, $\rho(x, x+\delta x) = e^{-\frac{1}{2}\delta x^2 g}$ where the metric tensor $g$ is a constant here, then the fourth-order Taylor expansion in $\delta x$, of the Gaussian correlation, leads to the identity $\mathbb{E}\left[\varepsilon \partial_x^4 \varepsilon\right] = 3g^2$ which is independent of the position $x$. As a possible closure, this suggest to model the unkown term as $\mathbb{E}\left[\varepsilon_u \partial_x^4 \varepsilon_u\right] \sim 3g_u^2 - 2\partial_x^2 g_u$ that depends on $x$. Replacing $g_u$ by $1/(2\nu_u)$ leads to

$$\mathbb{E}\left[\varepsilon_u \partial_x^4 \varepsilon_u\right] \sim \frac{\frac{\partial^2}{\partial x^2}\nu_{u,xx}(t,x)}{\nu_{u,xx}{}^2(t,x)} + \frac{3}{4}\frac{1}{\nu_{u,xx}{}^2(t,x)} - 2\frac{\left(\frac{\partial}{\partial x}\nu_{u,xx}(t,x)\right)^2}{\nu_{u,xx}{}^3(t,x)}. \tag{A1}$$

It results that Eq. (A1) is not the true analytic expresion of $\mathbb{E}\left[\varepsilon_u \partial_x^4 \varepsilon_u\right]$ as a function of $u, V_u$ and $\nu_u$ but only a parameterizations.

*Author contributions.* OP and RF designed the study, conducted the analysis, and wrote the manuscript. OP developed the code.

*Competing interests.* The authors declare that they have no conflict of interest.

*Acknowledgements.* The UML class diagram has been generated from UMLlet Auer et al. (2003). This work was supported by the French national program LEFE/INSU (Étude du filtre de KAlman PAramétrique, KAPA). RF has been partially supported by Labex Cominlabs

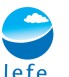

(grant SEACS), CNES (grant OSTST-MANATEE) and ANR through programs EUR Isblue, Melody and OceaniX.

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
