# Peer review of "PDE-NetGen 1.0: from symbolic PDE representations of physical processes to trainable neural network representations."

_Geoscientific Model Development, 2020_

## Referee Comment (RC1) · Anonymous Referee #1 · 2 Mar 2020

Although the authors suggest that the function closure may be modeled with deep learning architectures, no experiments in this direction are shown.

---

## Referee Comment (RC2) · Anonymous Referee #2 · 30 Mar 2020

This paper introduced the capabilities of the software package PDE-NetGen which the authors have developed. The main components of the software package are: 1) Finite-difference discretizations of spatial operators are implemented as convolutional layers with fixed parameters 2) Implementing time discretization methods as ResNet layers 3) Using the resulting network for forward or inverse problems

The authors show examples of how the framework can be used to derive NN implementations of right-hand side functions or more interestingly, the NN model can be used for data-driven parameter estimation of incertain models. In their examples, authors show

how a difficult-to-compute term in the equations can be modeled given some idea of its dependencies. The application is interesting and software is openly accessible for researchers. This in my opinion, warrants the paper to be published, however, in its current form the paper can benefit from a round of editing and improvements in the technical representation and writing. I have attached some highlights of the sentences that need revisiting, but I encourage the authors to consider a comprehensive refinement of the text.

Some suggestions: * The authors may include some information about the. orders of accuracy available in their software, especially what is the finite difference approximation? * Eq. (4) is very long. Is there a way to represent it in a matrix form to make it shorter? * On line 135 the authors intend to show that the solutions converge. The authors may consider supplementing the information with dt of the time-integrator so that the error has a sense of scale. * The sentence on line 175 is ambiguous. Do authors intend to say that the choice of a,b,c. = (1,3/4,-2) is not based on a ground truth? Interestingly their numerical solution somewhat close to this one. *There are misspellings highlighted in the pdf.

Please also note the supplement to this comment:
https://www.geosci-model-dev-discuss.net/gmd-2020-35/gmd-2020-35-RC2-supplement.pdf

**Supplement:**

[revised manuscript text omitted]

---

## Author Comment (AC1) · 23 Apr 2020

We would like to thank the referee for his/her comments. To answer to some of his/her points:

*P: One of the advantages of the convolutional layer translation is implementation in modern deep learning frameworks.*

A: We agree with the referee comment, and we exploit convolutional layers to fill the gap between partial differential and neural network. Introduced in a time-integration,

this produces an efficient implementation of the dynamics for the known part. Not that the connexion between neural network and differential equation has lead to some better understanding of what can be done by a neural network (ODE-Net, ResNet, bilinear layers,..) and is an active area of research. The package we propose, helps to fill the gap between the statistics and the physics in facilitating the development of useful architectures for evolution equations as encountered in geophysics.

*P: Although the authors suggest that the function closure may be modeled with deep learning architectures, no experiments in this direction are shown.*

A: The aim of the manuscript is not to introduce a deep learning architecture for the closure, but to facilitate the construction of a deep learning architecture taking into account the known physics: the focus is on the hybridation between the physics and the machine learning. Though the closure itself may not result in a deep architecture, the overall generated model leads to a deep architecture. More precisely, in the reported experiment, the number of layers introduced to train the closure is 6+5+13+25=49 layers for the known part of the dynamics and 2+3+2+4+4=15 layers for the neural network used in the closure. The ResNet implementation of the RK4 uses 11 layers. Hence, there are 75 layers used, with several convolutional layers among them. This is not a simple neural network, but at the end, it is able to learn from the data. The aim of this example was to focus on the neural network generation of the known part of the dynamics in order to facilitate the discovery of unknown terms, and we chose a simple problem to illustrate this. As discussed in Section 3.3 (p10-11), the implementation of an unknown term depends on the amount of knowledge we have. Here we chose to close the term from partial derivatives. For other problems, there would be no other choice than considering a deep neural network, for instance using multiple ResNet blocks, normalization, and so on, or architectures inspired from recent studies on closure modeling (eg, Bolton et al., 2019). And this can be plugged in our package as an exogenous neural network. Note that in the revision of the manuscript we will include an additional way to facilitate the design of the dynamics without plugging an

exogenous model – which we think to be easier for the physicist not used to handle neural network layers.

We start to prepare a revision of the manuscript considering his/her comments.

---

## Author Comment (AC2) · 23 Apr 2020

We would like to thank the referee for his/her comments. To answer to some of his/her questions:

*Q: The authors may include some information about the orders of accuracy available in their software*

A: The finite difference used here computes an approximation of any derivative at the second order of consistency (i.e. the approximation is equal to the true derivative plus

an error of order $O(dx^2)$ ). This will be made clearer in the revised version of the manuscript.

*Q: Do authors intend to say that the choice of $a, b, c = (1, 3/4, -2)$ is not based on a ground truth*

A: The closure is obtained when the local correlation function is approximated by a Gaussian correlation function. Hence, this relies on a theoretical ground but with an approximation for the correlation, so it is not the truth. An appendix has been prepared to explain this for the self-consistency of the manuscript.

We have a question concerning the comment (4):

*On line 135 the authors intend to show that the solutions converge. The authors may consider supplementing the information with dt of the time-integrator so that the error has a sense of scale.*

It is not clear to us, we understand the referee wants the value of $dt$, is it what he/she wants ? In our experiments, the value of $dt$ is set to 0.0016.

We start to prepare a revision of the manuscript considering his/her comments.
* * *

---

## Author Response (AR1)

Final answer to the referee 1

First of all, we would like to thank the referee for her/his review on our paper and for giving us the opportunity to improve our paper.

Now, we organized the answer to the comments as follows. First, we list some changes afford to the manuscript then detail our answers to the questions raised by the referee.

**List of changes for the revision**

*Major changes*

To facilitate the construction of a closure, we introduced the possibility to define an unknown quantity as a trainable scalar function using TrainableScalar class. This applies when the proposal for the closure is given by a symbolic expression with partial derivatives. Hence, there are now two possibilities to introduce a closure and merge the known physics to the design of a NN. The manuscript has been modified from lines 180 to 191, with a new Fig5 (see snapshot Rev1-Fig. 1)

*Minor changes*

1) Fig. 1, 2 and 6 has been updated to simplify the import of pdenetgen – this has no impact on the manuscript nor on the results.

2) The description of the learning was not precise, we modified the lines 205-210 as follows:
"Using an initial learning rate of 0.1, the training converges within 3 outer loops of 30 epochs with a geometrical decay of the learning rate by a factor of 1/10 after each outer loop."

*Differences between the two version of the manuscript*

To facilitate the comparison between the two version of the manuscript, a companion version of the manuscript lists all the modifications where old (new) statements are in red (blue). Snapshots of the revised version of manuscript help to illustrate the modifications, they are label as Rev1-Fig. X.

180    The above approach, which consists in constructing an exogenous function given by a NN to be determined, may seem tedious for an experimenter who would not be accustomed to NNs. Fortunately, we have considered an alternative in *PDE-*

**Introduction of the closure ine the PKF dynamics**

```python
from pdenetgen import TrainableScalar

**Set the closure by using TrainableScalar**
a, b, c = [TrainableScalar(l) for l in 'abc']
closure_proposal = a*Derivative(nu,x,2)/nu**Integer(2)+b*1/nu**Integer(2)+\
                   c*Derivative(nu,x)**2/nu**Integer(3)
display(closure_proposal)
```

$$\frac{a\frac{\partial^2}{\partial x^2} v_{u,xx}(t,x)}{v_{u,xx}{}^2(t,x)} + \frac{b}{v_{u,xx}{}^2(t,x)} + \frac{c\left(\frac{\partial}{\partial x} v_{u,xx}(t,x)\right)^2}{v_{u,xx}{}^3(t,x)}$$

```python
**Replace the closure(t,x) by the proposed closure**
pkf_dynamics[2] = pkf_dynamics[2].subs(Function('closure')(t,x),closure_proposal)

**Generate the NN code leading to the ClosedPKFBurgers class.**
exec(NNModelBuilder(pkf_dynamics,'ClosedPKFBurgers').code)
```

**Sample of code generated to define the ClosedPKFBurgers class**

```python
[..]
pow_21 = keras.layers.multiply([div_17,div_17,] ,name='PowLayer_21')
mul_28 = keras.layers.multiply([pow_21,Dnu_u_xx_x_o2],name='MulLayer_28')
train_scalar_9 = TrainableScalarLayerFactory(input_shape=mul_28.shape, name='TrainableScalar_a',
                 init_value=0,use_bias=False,mean=0.0,stddev=1.0,seed=None,wl2=None)(mul_28)
                 #TrainableScalar name: 'a'
add_8 = keras.layers.add([train_scalar_7,train_scalar_8,train_scalar_9],name='AddLayer_8')
mul_26 = keras.layers.multiply([pow_17,add_8],name='MulLayer_26')
[..]
```

**Figure 5.** Implementation of the closure, by defining each unknown quantity as an instance of the class *TrainableScalar*, and the resulting generated NN code. This is a part of code avaibale in the Jupyter notebook given as example in the package *PDE-NetGen*.

*NetGen* that can be used in the particular case where candidates for a closure take the form of an expression with partial derivatives, as it is the case for Eq. (8). An example of implementation is shown in Fig. 5 where *pkf_dynamics* stands for the system of equations Eq. (7). The unkown closure function is replaced by the proposal of closure Eq. (8) where each
185    unknown quantity $(a, b, c)$ is declared as an instance of the class *TrainableScalar*. Then, the NN is generated producing the class *ClosedPKFBurgers* whose an instance is ready for training. In the generated code, each instance of the *TrainableScalar* class is translated as a specific layer, *TrainableScalarLayerFactory*, equivalent to the above mentioned convolution layer, and whose parameter can be trainable. For instance, the trainable scalar $a$ is implemented by the line *train_scalar_9*. Note that the layer *TrainableScalarLayerFactory* can be used for 1D, 2D or 3D domains. In this example, the proposal for closure has been
190    defined at a symbolic level, without additional exogenous NN.

    An example of implementation for the exogenous NN and for the Trainable layers are provided in the package *PDE-NetGen* as Jupyter notebooks, for the case of the Burgers equation.

Rev1-Fig 1 : Introduction of the TrainableScalar class that facilitates the design of a NN when the unknown terms are given as symbolic expressions (lines 180-191).

**Answer to the point mentioned by the referee**

We copied your commentary in italics below, we reply in normal blue font.

*1. One of the advantages of the convolutional layer translation is implementation in modern deep learning frameworks.*

We agree with the referee comment and we exploit convolutional layers to bridge partial differential equations and neural networks. Introduced in a time-integration scheme, this produces an efficient implementation of the dynamics for the known part. We may point out that the connexion between neural networks and differential equations has led to some better understanding of some neural network architectures (ODE-Net, ResNet, bilinear layers,..) and is an active area of research. The package we propose, aims to make easier the exploitation of deep learning frameworks for physics (and vice versa) through a plug-an-play definition of useful NN architectures for evolution equations as encountered in geophysics.

*2. Although the authors suggest that the function closure may be modeled with deep learning architectures, no experiments in this direction are shown.*

The aim of the manuscript is not to introduce a deep learning architecture for the closure, but to facilitate the construction of a deep learning architecture taking into account the known physics: the focus is on the hybridation between physics and machine learning. Though the closure itself may not result in a deep architecture, the overall generated model leads to a deep architecture.

More precisely, in the reported experiment, the number of layers introduced to train the closure is 6+5+13+25=49 layers for the known part of the dynamics and 2+3+2+4+4=15 layers for the neural network used in the closure. The ResNet implementation of the RK4 uses 11 layers. Hence, there are 75 layers used, with several convolutional layers among them. In that sense, the overall model can be considered as a deep architecture, parameters of which can be learnt from data. The aim of this example was to focus on the neural network generation of the known part of the dynamics in order to facilitate the discovery of unknown terms, and we chose a simple problem to illustrate this. As discussed in Section 3.3 (p10-11), the implementation of an unknown term depends on the amount of knowledge we have. Here we considered a closure term from partial derivatives. For other problems, there would be no other choice than considering a deep neural network, for instance using multiple ResNet blocks, normalization, and so on, or architectures inspired from recent studies on closure modeling (eg, Bolton et al., 2019). Such architectures can be plugged in our package as an exogenous neural network. Note that in the revision of the manuscript we also include an additional way to facilitate the design of the dynamics without plugging an exogenous model using TrainableScalar class – which we think to be easier for the physicist not used to handle neural network layers.

We remind/precise the aim of the manuscript in the discussion in Section 3.3 in lines 232-240 (see Rev1-Fig2), considering the present answer to the referee.

The aim of the illustration proposed for the Burgers' dynamics is not to introduce a deep learning architecture for the closure, but to facilitate the construction of a deep learning architecture taking into account the known physics: the focus is

235  on the hybridation between physics and machine learning. Though the closure itself may not result in a deep architecture, the overall generated model leads to a deep architecture. For instance, the implementation using the exogenous NN use around 75 layers while the implementation based on the class *TrainableScalar* use 73 layers (we save the calculation of the derivatives that appear in Eq. (8), while they are computed twice when using the exogenous NN), with several convolutional layers among them. For other problems, there would be no other choice than considering a deep neural network, for instance using multiple

240  ResNet blocks, normalization, and so on, or architectures inspired from recent studies on closure modeling (*e.g.* , Bolton and Zanna (2019)). Such architectures can be plugged in *PDE-NetGen* as an exogenous neural network.

Rev1-Fig2 : Additional comment to precise the focus given in the manuscript.

Final answer to the referee 2

First of all, we would like to thank the referee for her/his review on our paper and for giving us the opportunity to improve our paper.

Now, we organized the answer to the comments as follows. First, we list some changes afford to the manuscript then detail our answers to the questions raised by the referee.

**List of changes for the revision**

*Major changes*

To facilitate the construction of a closure, we introduced the possibility to define an unknown quantity as a trainable scalar function using TrainableScalar class. This applies when the proposal for the closure is given by a symbolic expression with partial derivatives. Hence, there are now two possibilities to introduce a closure and merge the known physics to the design of a NN. The manuscript has been modified from lines 180 to 191, with a new Fig5 (see snapshot Rev2-Fig. 1)

*Minor changes*

1) Fig. 1, 2 and 6 has been updated to simplify the import of pdenetgen – this has no impact on the manuscript nor on the results.

2) The description of the learning was not precise, we modified the lines 205-210 as follows:
"Using an initial learning rate of 0.1, the training converges within 3 outer loops of 30 epochs with a geometrical decay of the learning rate by a factor of 1/10 after each outer loop."

*Differences between the two version of the manuscript*

To facilitate the comparison between the two version of the manuscript, a companion version of the manuscript lists all the modifications where old (new) statements are in red (blue). Snapshots of the revised version of manuscript help to illustrate the modifications, they are label as Rev2-Fig. X.

180     The above approach, which consists in constructing an exogenous function given by a NN to be determined, may seem tedious for an experimenter who would not be accustomed to NNs. Fortunately, we have considered an alternative in *PDE-*

Introduction of the closure ine the PKF dynamics

```
from pdenetgen import TrainableScalar

**Set the closure by using TrainableScalar**
a, b, c = [TrainableScalar(l) for l in 'abc']
closure_proposal = a*Derivative(nu,x,2)/nu**Integer(2)+b*1/nu**Integer(2)+\
                c*Derivative(nu,x)**2/nu**Integer(3)
display(closure_proposal)
```

$$a \frac{\frac{\partial^2}{dx^2} v_{u,xx}(t,x)}{v_{u,xx}^2(t,x)} + \frac{b}{v_{u,xx}^2(t,x)} + \frac{c\left(\frac{\partial}{dx} v_{u,xx}(t,x)\right)^2}{v_{u,xx}^3(t,x)}$$

```
**Replace the closure(t,x) by the proposed closure**
pkf_dynamics[2] = pkf_dynamics[2].subs(Function('closure')(t,x),closure_proposal)

**Generate the NN code leading to the ClosedPKFBurgers class.**
exec(NNModelBuilder(pkf_dynamics,'ClosedPKFBurgers').code)
```

Sample of code generated to define the ClosedPKFBurgers class

```
[..]
pow_21 = keras.layers.multiply([div_17,div_17,] ,name='PowLayer_21')
mul_28 = keras.layers.multiply([pow_21,Dnu_u_xx_x_o2],name='MulLayer_28')
train_scalar_9 = TrainableScalarLayerFactory(input_shape=mul_28.shape, name='TrainableScalar_a',
                init_value=0,use_bias=False,mean=0.0,stddev=1.0,seed=None,wl2=None)(mul_28)
                #TrainableScalar name: 'a'
add_8 = keras.layers.add([train_scalar_7,train_scalar_8,train_scalar_9],name='AddLayer_8')
mul_26 = keras.layers.multiply([pow_17,add_8],name='MulLayer_26')
[..]
```

**Figure 5.** Implementation of the closure, by defining each unknown quantity as an instance of the class *TrainableScalar*, and the resulting generated NN code. This is a part of code avaibale in the Jupyter notebook given as example in the package *PDE-NetGen*.

*NetGen* that can be used in the particular case where candidates for a closure take the form of an expression with partial derivatives, as it is the case for Eq. (8). An example of implementation is shown in Fig. 5 where *pkf_dynamics* stands for the system of equations Eq. (7). The unkown closure function is replaced by the proposal of closure Eq. (8) where each

185     unknown quantity $(a, b, c)$ is declared as an instance of the class *TrainableScalar*. Then, the NN is generated producing the class *ClosedPKFBurgers* whose an instance is ready for training. In the generated code, each instance of the *TrainableScalar* class is translated as a specific layer, *TrainableScalarLayerFactory*, equivalent to the above mentioned convolution layer, and whose parameter can be trainable. For instance, the trainable scalar $a$ is implemented by the line *train_scalar_9*. Note that the layer *TrainableScalarLayerFactory* can be used for 1D, 2D or 3D domains. In this example, the proposal for closure has been

190     defined at a symbolic level, without additional exogenous NN.

    An example of implementation for the exogenous NN and for the Trainable layers are provided in the package *PDE-NetGen* as Jupyter notebooks, for the case of the Burgers equation.

Rev2-Fig 1 : Introduction of the TrainableScalar class that facilitates the design of a NN when the unknown terms are given as symbolic expressions (lines 180-191).

**Answer to the question of the referee**

We copied your commentary in italics below, we reply in normal blue font.

*1. "The authors may include some information about the orders of accuracy available in their software, especially what is the finite difference approximation?"*

In order to explain what is the finite-difference method, we have modified the manuscript as

follows:
- ৺ we add a reference to a book (Thomas, 1995) (in the introduction at line 32).
- ৺ we shortly explain what is the basic idea of the method through the sentence "The finite-difference method remains to replace the derivative of a function by a fraction that only depends on the value of the function (see e.g. Thomas, 1995)." (in Section 2.1, lines 58-59)
- ৺ we detail the computation of the finite-difference approximation of a derivative as implemented in PDE-NetGen. In particular we detail that the finite-difference implementation is consistent at the second order (see Section 2.1, lines 69-77).

**2.1 Introducing physical knowledge in the design of a NN topology**

55  Since the NN generator is designed for evolution equations, the core of the generator is the automatic translation of partial derivatives with respect to spatial coordinates into layers. The correspondence between the finite-difference discretization and the convolutional layer give a practical way to translate a PDE into a NN (Cai et al., 2012; Dong et al., 2017; Long et al., 2017).

The finite-difference method remains to replace the derivative of a function by a fraction that only depends on the value of the function (see *e.g.* Thomas (1995)). For instance, the finite-difference method applied on a second order partial derivative

60  $\partial_x^2 u$, for $u(t,x)$ on a 1D domain, leads to approximate the derivative by

$$\partial_x^2 u(t,x) \approx \mathcal{F}_x^2 u(t,x), \tag{2}$$

with

$$\mathcal{F}_x^2 u(t,x) = \frac{u(t,x+\delta x) + u(t,x-\delta x) - 2u(t,x)}{\delta x^2}, \tag{3}$$

where $\delta x$ stands for the discretization space step. Here the spatial derivative is replaced by a fraction that only depends on the

65  values of $u$ at the time $t$ and points $x - \delta x$, $x$, $x + \delta x$. This makes appear a kernel stencil $k = [1/\delta x^2, -2/\delta x^2, 1/\delta x^2]$ that can be used in a 1D convolution layer with a linear activation function and without bias. A similar routine applies for 2D and 3D geometries. *PDE-NetGen* relies on the computer algebra system *sympy* (Meurer et al., 2017) to compute the stencil as well as to handle symbolic expressions.

In *PDE-NetGen*, the finite-difference implementation appears as a linear operator $\mathcal{F}$ which approximates any partial deriva-

70  tive from the values on a regular grid. In particular, the finite difference $\mathcal{F}_x^\alpha u(t,x)$ of any partial derivative $\partial_x^\alpha u(t,x)$ of order $\alpha$, is computed from the grid points $\{x \pm (2i+1)\delta x\}_{i\in[0,p]}$ when $\alpha = 2p+1$ is odd and $\{x \pm i\delta x\}_{i\in[0,p]}$ when $\alpha = 2p$ is even. This approximation is consistent at the second order *i.e.* $\mathcal{F}_x^\alpha u \underset{0}{=} \partial_x^\alpha u + \mathcal{O}(\delta x^2)$, where $\mathcal{O}$ is the Landau's big O notation: for any $f$, the notation $f(\delta x) \underset{0}{=} \mathcal{O}(\delta x^2)$ means that $\lim_{\delta x \to 0} \frac{f(\delta x)}{\delta x^2}$ is finite. The operator $\mathcal{F}$ behaves partially as the partial derivative operator $\partial$: $\mathcal{F}$ is commutative with respect to independent coordinates *i.e.* in a 2D domain for coordinates $(x,y)$

75  we have $\mathcal{F}_x \circ \mathcal{F}_y = \mathcal{F}_y \circ \mathcal{F}_x$, where $\circ$ denotes the operator composition, and this applies at any order *e.g.* $\mathcal{F}_{xxy}^3 = \mathcal{F}_x^2 \circ \mathcal{F}_y$ (but $\mathcal{F}_x^2 \neq \mathcal{F}_x \circ \mathcal{F}_x$). Hence, the finite difference of a derivative with respect to multiple coordinate, is computed sequentially from the iterative discretization along each coordinate, and this approximation is consistent at the second order. Note that we chose to design *PDE-NetGen* considering the finite-difference method, but alternatives using automatic differentiation can be considered as introduced by Raissi (2018) who used TensorFlow for the computation of derivative.

80  Then, the time integration can be implemented either by a solver or by a ResNet architecture of a given time scheme *e.g.* an Euler scheme or a fourth order Runge-Kutta (RK4) scheme (Fablet et al., 2017).

These two components, namely the translation of partial derivatives into NN layers and a ResNet implementation of the time integration, are the building blocks of the proposed NN topology generator as examplified in the next Section.

Rev2-Fig 2 : Details of the finite-difference method as implemented in PDE-NetGen.

*2. "Eq. (4) is very long. Is there a way to represent it in a matrix form to make it shorter?"*

This has been replaced by the much simpler equation Eq.(5), where we used the description of the finite-difference method as implemented in the package. See the modification on the snapshot Rev2-Fig 3.

a fourth-order Runge-Kutta scheme. The initial condition of the simulation is given by a Dirac Fig. 4 (a). In order to validate the solution obtained from the generated neural network, we compare the integration with the one of the finite-difference discretization of Eq. (4),

$$\quad \partial_t u = \mathcal{F}_{x^i}(\kappa_{ij})\mathcal{F}_{x^j}(u) + \kappa_{ij}\mathcal{F}^2_{x^i x^j}(u), \tag{5}$$

where $\mathcal{F}$ is the operator described in Section 2.1, and whose numerical result is shown in Fig. 4 (b).

Rev2-Fig 3 : New equation for the finite-difference discretization of the diffusion equation.

*3. "On line 135 the authors intend to show that the solutions converge. The authors may consider supplementing the information with dt of the time-integrator so that the error has a sense of scale."*

The time step has been added, $dt = 1.66 \, 10^{-3}$.

*4. "The sentence on line 175 is ambiguous. Do authors intend to say that the choice of a,b,c. = (1,3/4,-2) is not based on a ground truth? Interestingly their numerical solution somewhat close to this one."*

The Appendix A has been introduced to provide some details on the closure term considered in this experiment. In particular, this explains why this is not the truth (see Rev2-Fig 4). For the configuration used here, the theoretical closure is accurate, and we wanted to test if a neural network was able to reproduce this, which is the case following the results we present. The manuscript has been modified to include the reference to Appendix A (see Rev2-Fig 5)

**Appendix A: Local Gaussian closure**

260  For self-consistency, we detail how the theoretical closure is obtained (Pannekoucke et al., 2018).

It can be shown that $\mathbb{E}\left[\varepsilon_u \partial_x^4 \varepsilon_u\right] = \mathbb{E}\left[\left(\partial_x^2 \varepsilon_u\right)^2\right] - 2\partial_x^2 g_u$ where $g_u = \frac{1}{2\nu_u}$ is the so-called metric tensor that is a scalar field in 1D. When the correlation function $\rho(x, x+\delta x) = \mathbb{E}\left[\varepsilon(x)\varepsilon(x+\delta x)\right]$ is a homogeneous Gaussian, $\rho(x, x+\delta x) = e^{-\frac{1}{2}\delta x^2 g}$ where the metric tensor $g$ is a constant here, then the fourth-order Taylor expansion in $\delta x$, of the Gaussian correlation, leads to the identity $\mathbb{E}\left[\varepsilon\partial_x^4\varepsilon\right] = 3g^2$ which is independent of the position $x$. As a possible closure, this suggest to model the unkown

265  term as $\mathbb{E}\left[\varepsilon_u\partial_x^4\varepsilon_u\right] \sim 3g_u^2 - 2\partial_x^2 g_u$ that depends on $x$. Replacing $g_u$ by $1/(2\nu_u)$ leads to

$$\mathbb{E}\left[\varepsilon_u\partial_x^4\varepsilon_u\right] \sim \frac{\frac{\partial^2}{\partial x^2}\nu_{u,xx}(t,x)}{\nu_{u,xx}^2(t,x)} + \frac{3}{4}\frac{1}{\nu_{u,xx}^2(t,x)} - 2\frac{\left(\frac{\partial}{\partial x}\nu_{u,xx}(t,x)\right)^2}{\nu_{u,xx}^3(t,x)}. \tag{A1}$$

It results that Eq. (A1) is not the true analytic expresion of $\mathbb{E}\left[\varepsilon_u\partial_x^4\varepsilon_u\right]$ as a function of $u$, $V_u$ and $\nu_u$ but only a parameterizations.

Rev2-Fig 4 : New Appendix that provides details on the closure

tion 2. This expression is similar to a dictionary of possible terms as in Rudy et al. (2017) and it is inspired from an arbitrary

175   theoretically-designed closure for this problem where $(a, b, c) = (1, \frac{3}{4}, -2)$ (see Appendix A for details). In the NN implementation of the exogenous function modeled as Eq. (8), each of the unknown coefficients $(a, b, c)$ are implemented as a 1D

convolutional layer, with a linear activation function and without bias. Note that the estimated parameters $(a, b, c)$ could be different from the one of the theoretical closure: while the theoretical closure can give some clues for the design of the unknown

term, this closure is not the truth which is unknown (see Appendix A).

Rev2-Fig 5 : Modification of the part concerning the closure.

*5. "There are misspellings highlighted in the pdf."*

*The manuscript has been modified, thank you very much!*

[revised manuscript text omitted]